# REVISIT THE OPEN NATURE OF OPEN VOCABULARY SEMANTIC SEGMENTATION

**Qiming Huang, Han Hu, Jianbo Jiao**
The MIx Group, School of Computer Science
University of Birmingham
{qxh366, hxh864}.student@bham.ac.uk, j.jiao@bham.ac.uk

## ABSTRACT

In Open Vocabulary Semantic Segmentation (OVS), we observe a consistent drop in model performance as the query vocabulary set expands, especially when it includes semantically similar and ambiguous vocabularies, such as *'sofa'* and *'couch'*. The previous OVS evaluation protocol, however, does not account for such ambiguity, as any mismatch between model-predicted and human-annotated pairs is simply treated as incorrect on a pixel-wise basis. This contradicts the open nature of OVS, where ambiguous categories may both be correct from an open-world perspective. To address this, in this work, we study the open nature of OVS and propose a mask-wise evaluation protocol that is based on matched and mismatched mask pairs between prediction and annotation respectively. Extensive experimental evaluations show that the proposed mask-wise protocol provides a more effective and reliable evaluation framework for OVS models compared to the previous pixel-wise approach on the perspective of open-world. Moreover, analysis of mismatched mask pairs reveals that a large amount of ambiguous categories exist in commonly used OVS datasets. Interestingly, we find that reducing these ambiguities during both training and inference enhances capabilities of OVS models. These findings and the new evaluation protocol encourage further exploration of the open nature of OVS, as well as broader open-world challenges. Project page: https://qiming-huang.github.io/RevisitOVS/.

> "The limits of my language mean the limits of my world."
>
> *Ludwig Wittgenstein*

## 1 INTRODUCTION

Open-world learning aims to address the problem of learning with novel and unknown categories or data distributions that are often encountered in the real world. With the development of large language-vision (LLV) models, such as CLIP (Radford et al., 2021), open vocabulary tasks are proposed to utilise the strong language visual alignment capability from LLV to identify objects from the data, including new entities not in the training data. Particularly, open vocabulary semantic segmentation (OVS) is a task where models trained on a close-set semantic segmentation dataset perform zero-shot inference on an unseen dataset by providing a vocabulary set for any object.

In an open-world setting, object category boundaries are often ambiguously defined. For instance, when describing visual objects through language, highly ambiguous vocabulary may be used. While this works in closed-set settings, where labels are assumed to be mutually exclusive. However, in the case of open vocabulary semantic segmentation (OVS), where any vocabulary, including those with significant semantic ambiguity, can be introduced. For example, as illustrated in Fig. 1, under the current OVS evaluation protocol, predictions such as *'flower'* and *'chair'* are considered incorrect, despite appearing reasonable to humans from an open-world perspective. In this paper, we aim to address the challenges posed by such ambiguous categories by revisiting the *open nature* of OVS, trying to answer: *whether we should treat the ambiguous categories as incorrect or correct ones, and how to encourage OVS to be more open?*

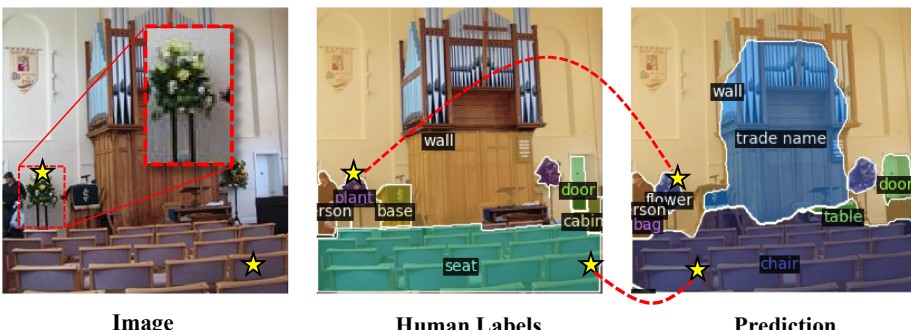

**Image**        **Human Labels**        **Prediction**

Figure 1: Category ambiguity in open vocabulary semantic segmentation. One object can be assigned multiple possible labels while the human label is only one of them. For example, the area on the left with a yellow star was annotated as *'plant'* by humans, but predicted to be *'flower'* by the OVS model; the bottom part annotated as *'seat'* was predicted as *'chair'* by OVS model.

Specifically, we first revisit the existing OVS evaluation process from the perspective of open-world learning, and find that it follows a closed-set approach, where predictions are considered incorrect if they do not match with the predefined category. We observe that as the number of inference categories increases, the performance of existing OVS models significantly declines, indicating issues caused by category ambiguity. To study this, we find that expanding model predictions from pixel-wise argmax to category-wise mask-wise predictions effectively mitigates such ambiguity problems. To this end, we propose an open-set prediction approach and a corresponding generalised category-preserving evaluation metric. We find that the proposed new prediction and evaluation framework significantly improves the performance of existing OVS methods. Additionally, based on our proposed evaluation framework, we construct an ambiguous vocabulary graph between model predictions and human annotations, revealing clear community structures where vocabularies within the same community correspond to visually similar objects.

The main contributions of this study can be summarised as follows: 1) We revisit existing OVS paradigms from an open-world perspective, offer insightful observations, and propose feasible solutions to encourage more openness in OVS. 2) Our proposed mask-wise evaluation protocol effectively addresses the issue of ambiguous categories in open-world evaluation. 3) Based on the proposed evaluation framework, we introduce an approach to conltruct a confusion vocabulary graph for existing OVS datasets, highlighting the significant presence of ambiguous category annotations. 4) Extensive experimental analysis and comparisons validate the effectiveness of the proposed method, which achieves state-of-the-art performance with a new paradigm for OVS in the *open* world.

## 2 RELATED WORKS

### 2.1 CLOSE-SET SEGMENTATION

This task aims to segment images into regions with predefined categories. Fully Convolutional Networks (FCNs) (Long et al., 2015) marked the beginning of the deep learning era in image segmentation. Subsequently, convolution-based (Li et al., 2023) and Transformer-based (Liu et al., 2021) approaches further enhanced the model's performance in semantic segmentation (Zhao et al., 2017; Wang et al., 2020; Shen et al., 2022; Chen et al., 2017; Ronneberger et al., 2015; Xie et al., 2021), instance segmentation (He et al., 2017; Lin et al., 2014; Chen et al., 2019a; Liu et al., 2018; Qi et al., 2021), and panoptic segmentation (Kirillov et al., 2019b;a; Cheng et al., 2021; Zhang et al., 2021). Despite the continuous advancements in closed-set segmentation methods, predefined category sets are inadequate for open-world vision applications where the number of object categories is vast and constantly evolving. In this work, we focus on adapting closed-set evaluation metrics to accommodate open-world scenarios, especially for dealing with the case of ambiguous categories.

## 2.2 OPEN VOCABULARY SEMANTIC SEGMENTATION

Close-set segmentation aims to train the model to segment predetermined categories while OVS aims to segment the objects with arbitrary vocabulary queries (Cho et al., 2023; Xie et al., 2023; Xu et al., 2023) which enables the open ability for model prediction. There has been some recent exploratory work in this direction. LSeg (Li et al., 2022) utilised CLIP (Radford et al., 2021) to train a visual encoder that generates pixel-level visual embeddings from an image, which are aligned with the corresponding textual embeddings learned from the training labels within the CLIP embedding space. OpenSeg (Ghiasi et al., 2022) employed a class-agnostic segmentation module utilising region-to-image cross-attention to detect local regions in images. Two-stage frameworks, ZegFormer (Ding et al., 2022) and ZSseg (Xu et al., 2022), also extract class-agnostic region proposal similar to (Ghiasi et al., 2022) at the first stage, then utilised pretrained vision-language models like CLIP to classify masked regions. Liang et al. (2023) improves CLIP's performance on masked images by finetuning CLIP on image-text pairs. CAT-Seg (Cho et al., 2023) proposed a cost aggregation method to optimise the image-text similarity map by fine-tuning the CLIP encoder and obtaining accurate pixel-level predictions. MaskCLIP (Dong et al., 2023) integrates a novel masked self-distillation technique into contrastive language-image pretraining, aiming to derive pixel-level embeddings from CLIP for immediate application in segmentation tasks. SED (Xie et al., 2023) introduced an encoder-decoder architecture consisting of a hierarchical encoder-based cost map generation and a gradual fusion decoder with category early rejection to obtain pixel-level image-text cost map prediction. Despite the progress achieved by the previous OVS models, the existing training and inference of OVS still adhere to the pipeline of close-set recognition, *i.e.* a fixed training vocabulary set is used during training, and a particular dataset-specific vocabulary set is given during inference.

## 2.3 THE EVALUATION OF OVS

The current evaluation methods for open vocabulary semantic segmentation primarily rely on the mean Intersection over Union (mIoU) metric, which assesses classification accuracy at the pixel level. However, this approach depends on strict pixel matching, where predictions are deemed correct only if they match the ground truth class exactly, making it unsuitable for open-world scenarios. Recent studies (Zhou et al., 2023; Liu et al., 2023) attempt to address this by incorporating textual similarity between categories, assigning partial accuracy scores based on the degree of similarity during IoU computation. While this approach introduces flexibility, textual similarity alone cannot reliably capture the visual-semantic relationships between objects, limiting its effectiveness. In this work, we focus on evaluating models based on visual similarity, directly comparing the overlap between segmentation predictions and human annotations to better distinguish visually similar classes in open-world settings.

## 3 REVISITING OPEN VOCABULARY SEMANTIC SEGMENTATION

**Definition of OVS.** Open vocabulary semantic segmentation (OVS) aims to train segmentation models that can leverage textual descriptions to segment arbitrary objects. Given two category sets, $C_{train}$ and $C_{test}$, where $C_{train}$ and $C_{test}$ are not equal in terms of object categories ($C_{train} \neq C_{test}$), the model is trained on $C_{train}$ and directly tested on $C_{test}$. Typically, $C_{train}$ and $C_{test}$ are described using noun phrases (*e.g.* sky, ocean, mountains, *etc.*). During the testing stage, the previous assumption held by OVS is that it is known which dataset the test data originates from, and the whole vocabulary set corresponding to the dataset is provided as the inference vocabulary, $C_{D_1}$, $C_{D_2}, \ldots, C_{D_n}$.

**Training objective.** Consider the Maximum A Posterior (MAP) estimate for training a deep learning model with given data $X$ and a certain vocabulary set $\mathcal{V}$, let the prior distribution of $\Theta$ be $g(\Theta)$. We want to find a parameter $\Theta$ that maximises:

$$\Theta_{MAP} = \arg\max_{\Theta} P(\Theta|X, \mathcal{V}). \tag{1}$$

Since the training vocabulary set $\mathcal{V}$ is considered a fixed set in most previous OVS works (Cho et al., 2023; Xie et al., 2023; Xu et al., 2023), we apply Bayes' theorem to get:

$$\Theta_{MAP} \propto \arg\max_{\Theta} \ \log P_{\mathcal{V}}(X|\Theta) + \log g(\Theta). \quad (2)$$

Although OVS intends to incorporate vocabulary during training, the objective above fails to consider the vocabulary during the model optimisation. Here we propose to turn the training vocabulary into a random variable represented by $P(\mathcal{V})$, as aforementioned, we have our MAP estimate for training an OVS model:

$$
\begin{aligned}
\hat{\Theta}_{MAP} &= \arg\max_{\Theta} P(\Theta|X, \mathcal{V}) \\
&= \arg\max_{\Theta} \frac{P(X|\Theta, \mathcal{V})P(\mathcal{V}|\Theta)P(\Theta)}{P(X|\mathcal{V})P(\mathcal{V})} \\
&\propto \arg\max_{\Theta} \underbrace{\log P(X|\Theta, \mathcal{V})}_{likelihood} + \underbrace{\log P(\mathcal{V}|\Theta)}_{language\ likelihood} + \underbrace{\log P(\Theta)}_{prior}. \quad (3)
\end{aligned}
$$

We notice that compared to the original objective in Eq. 2, the current optimisation incorporates another term $P(\mathcal{V}|\Theta)$ that relates to the vocabulary distribution (for a detailed mathematical derivation, please refer to the supplementary material). Considering vocabulary $\mathcal{V}$ as a random variable during training, the model parameters $\Theta$ are dependent on both the observed training data $X$ and the vocabulary $\mathcal{V}$ through maximising the $P(\mathcal{V}|\Theta)$ and $P(X|\Theta, \mathcal{V})$ terms. This implicitly constructs the relationship between model parameters and vocabulary distribution, and could be beneficial to OVS in open world scenarios.

**Zero-shot inference capability.** Open vocabulary semantic segmentation models can perform zero-shot inference on unseen datasets while providing customisable vocabulary. Given an image $I_i \in \mathcal{R}^{B,C,W,H}$ and vocabulary candidate set $\mathcal{V} \in \mathcal{R}^{B,D}$, the OVS model takes $I_i$ and $\mathcal{V}$ as input, and generates a class posterior $P(y_i|X, \Theta, \mathcal{V})$. Usually the consequent semantic segmentation predictions are obtained by applying the argmax operation:

$$\hat{Y} = \arg\max_{y_i \in \mathcal{V}} P(y_i|X, \Theta, \mathcal{V}). \quad (4)$$

In zero-shot inference, a spatial posterior $\hat{Y} \in \mathbb{R}^{C,W,H}$ is generated, where $C$ denotes the number of classes, similar to the vocabulary in open vocabulary semantic segmentation (OVS). Each pixel is assigned to the class in $\mathcal{V}_{test}$ with the highest posterior probability. Existing OVS methods assume that the test images come from a specific dataset (*e.g.* ADE20K (Zhou et al., 2019) or PASCAL-Context (Mottaghi et al., 2014)), and they use the corresponding dataset-specific vocabulary to restrict all predictions to the dataset's vocabulary.

**The open nature of OVS.** In an open-world scenario, the open nature of visual semantic objects implies that they may belong to multiple labels (described by different vocabularies or captions), which is overlooked. This open nature in vocabulary segmentation emphasises that object classification is not rigid or strictly predefined, allowing for greater flexibility in category assignment and interpretation. Specifically, it encompasses two key aspects: 1) *Multiple Labels*: An object can simultaneously be associated with multiple category labels, reflecting the complexity and richness of real-world concepts. 2) *Semantic Similarity*: A predicted category may not exactly match the ground truth label, but if it is semantically similar, it should not be considered entirely incorrect. For example, predicting "vehicle" instead of "car" is still valid, as both vocabularies capture the essence of the object within a broader semantic context.

## 4 MASK-WISE EVALUATION PROTOCOL

### 4.1 NOTATIONS AND DEFINITIONS

For the sake of clear understanding, we first define the main symbols used in the proposed evaluation framework and their meanings. For image $\mathcal{X}_i$ from a testing dataset $\mathcal{D}$, we have the class probability

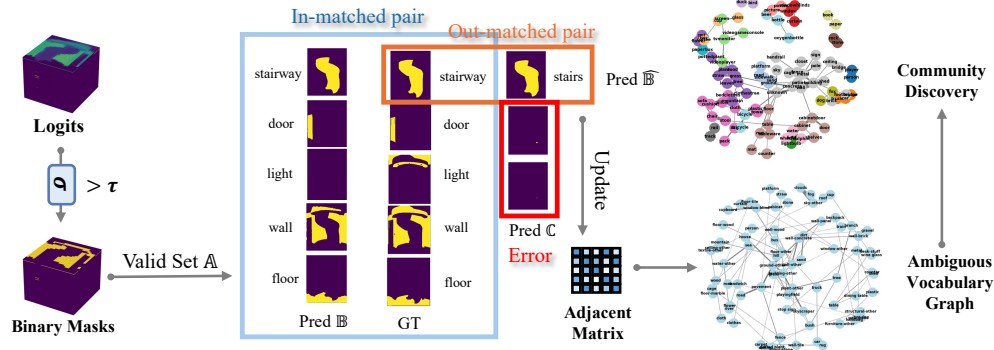

Figure 2: The proposed mask-wise evaluation protocol. The Valid Set $\mathbb{A}$ consists of all masks where $\mathbb{A} = \{M_i \mid M_i \in \{M_1, M_2, \ldots, M_K\}\}$. $\mathbb{B}$ represents the list of masks where the predicted category matches the category annotated in the ground truth (GT). $\hat{\mathbb{B}}$ is the set of masks obtained by performing bipartite matching between $(\mathbb{A} \setminus \mathbb{B})$ and the GT, where the IoU of the matched pairs exceeds the threshold $\tau_{AV}$. For example, *"stairs"* belongs to $\hat{\mathbb{B}}$ in this figure. $\mathbb{C}$ is defined as $\mathbb{C} = \mathbb{A} \setminus (\hat{\mathbb{B}} \cup \mathbb{B})$. The adjacent matrix helps construct the ambiguous vocabulary graph, enabling community discovery for a better understanding of model predictions. See Section 4.3 for details.

distribution predicted by the model, referred to as logits, denoted by $\mathbf{L}_i \in \mathbb{R}^{C \times W \times H}$, where $C$ represents the total number of classes, and $W$ and $H$ denote the width and height of the image, respectively. Pixel-wise semantic segmentation annotations are represented as $\mathbf{Y}_i \in \{0, 1, \ldots, C-1\}^{W \times H}$, where each value corresponds to the class index of the respective pixel. The one-hot encoded form of these annotations is given by $\mathbf{M}_i \in \{0, 1\}^{C \times W \times H}$, where each pixel is either 0 or 1, $\mathbf{M}_i^{true} \in \{0, 1\}^{C_{true} \times W \times H}$ is the mask list with all annotated category mask only, $C_{true} < C$. Binary predicted masks, $\mathbf{B}(\tau) \in \{0, 1\}^{C \times W \times H}$, are obtained by thresholding $\mathbf{L}_i$ with a threshold $\tau$, such that:

$$\mathbf{B}(\tau)_i = \begin{cases} 1, & \text{if } \mathbf{L}_i \geq \tau, \\ 0, & \text{otherwise.} \end{cases} \tag{5}$$

The $\mathbf{CM} \in \mathbb{N}^{C \times 2 \times 2}$ is a binary confusion matrix for dataset $\mathcal{D}$, where each class $c$ has:

$$\mathbf{CM}_c = \begin{bmatrix} \text{TP}_c & \text{FP}_c \\ \text{FN}_c & \text{TN}_c \end{bmatrix}. \tag{6}$$

$\mathbf{EM} \in \mathbb{R}^{C \times 1}$ is a vector used to quantify the error proportion for each class in the model's predictions. The error proportion is calculated as the ratio of the number of pixels with a value of 1 in the corresponding binary mask to the total number of pixels in the image $i$ as follows:

$$err_{i,c} = \frac{\text{Number of pixels with a value of 1 in the binary mask}}{\text{Total number of pixels}}. \tag{7}$$

## 4.2 EVALUATION PROTOCOL

Our mask-wise evaluation protocol provides CM, AV, and EM under different thresholds as detailed in Appendix A.1. Here, CM represents the binary confusion matrix between the predicted categories and the annotated categories. AV captures predictions with high overlap ($> \tau_{AV}$) between the predicted and annotated masks but with mismatched categories. Only CM and EM are used to evaluate the model's performance, as they are based on comparisons with the ground-truth annotations. AV, however, is utilised for subsequent ambiguous vocabulary graph analysis (*i.e.* analysing ambiguous vocabulary in the model's predictions) rather than directly assessing the model's performance.

Based on our proposed evaluation protocol, we define three metrics $front$, $back$ and $err$ to evaluate OVS model performance across different thresholds, defined as:

$$front_\tau = \frac{1}{|C|} \sum_{c \in C} \frac{\mathbf{CM}_\tau[TP_c]}{\mathbf{CM}_\tau[TP_c] + \mathbf{CM}_\tau[FP_c] + \mathbf{CM}_\tau[FN_c]} \tag{8}$$

$$back_\tau = \frac{1}{|C|} \sum_{c \in C} \frac{\mathbf{CM}_\tau[TN_c]}{\mathbf{CM}_\tau[TN_c] + \mathbf{CM}_\tau[FP_c] + \mathbf{CM}_\tau[FN_c]} \tag{9}$$

$$err_\tau = \frac{1}{|C|} \sum_{c \in C} \mathbf{EM}_{\tau,c} \tag{10}$$

Here, $front_{\tau,c}$ and $back_{\tau,c}$ represent the recognition IoU of the foreground and background classes, respectively, for each category $c$, where $c \in C$ under threshold $\tau$. The foreground class refers to the pixels labelled as relevant to the target in category $c$ (*i.e.* the regions belonging to that category), the background class refers to the pixels labelled as irrelevant to the target in category $c$ (*i.e.* regions outside that category). The $err_\tau$ represents the average proportion of incorrectly predicted pixels across all categories under the threshold $\tau$. Inspired by best F1 thresholding (Lipton et al., 2014), the best threshold $\tau^\star$ can be automatically determined by:

$$\tau^\star = \underset{\tau \in \{0.1, 0.2, \dots, 0.9\}}{\arg\max} \left( \sqrt{front_i^2 + (1 - err_i)^2} \right). \tag{11}$$

### 4.3 Ambiguous Graph in Out-Matched Pair

**Building ambiguous vocabulary graph.** Graph is a structure used to model pairwise relationships between objects. It is commonly described using an adjacency matrix, where each entry in the matrix represents the connection or interaction between two nodes (in this case, vocabularies). The confusion graph is constructed based on the model's predictions and the manually annotated classes. The graph is used for analysing the model's performance in classification task (Jin et al., 2017).

The adjacency matrix $\mathbf{AV}$ for ambiguous vocabulary graph is the ambiguous vocabulary matrix in Appendix A.1. Each element $\mathbf{AV}_{i,j}$ represents the number of times the model predicts class $j$, while the ground truth is class $i$. For each out-matched pair, the adjacency matrix is updated. For example, if there is an out-matched pair such as *"couch"*-*"sofa"* where the ground truth class is *"sofa"* and the predicted class is *"couch"*, we update the corresponding entry in the adjacency matrix, $\mathbf{AV}_{\text{sofa,couch}}$, by incrementing it by 1. This represents the frequency with which the model misclassified "sofa" as "couch" with a high IoU overlap (IoU $> \hat{\tau}$).

**Community discovery over ambiguous graph.** Given the confusion graph represented by the corresponding adjacency matrix $\mathbf{A} \in \mathbb{R}^{C \times C}$, we can perform community discovery to identify groups of classes that are frequently confused with each other. This process involves partitioning the nodes (classes) into communities such that nodes within the same community have stronger connections, as reflected by higher values in the adjacency matrix, than those across different communities.

One common approach for community discovery is modularity maximisation. The modularity $Q$ of a given partition of the graph is defined as:

$$Q = \frac{1}{2m} \sum_{i,j} \left[ A_{i,j} - \frac{k_i k_j}{2m} \right] \delta(c_i, c_j) \tag{12}$$

where $A_{i,j}$ is the weight of the edge between nodes $i$ and $j$ in the adjacency matrix, $k_i$ and $k_j$ are the degrees (total edge weights) of nodes $i$ and $j$, respectively, $m$ is the total weight of all edges in the graph, i.e. $m = \frac{1}{2} \sum_{i,j} A_{i,j}$. $\delta(c_i, c_j)$ is the Kronecker delta function, which is 1 if nodes $i$ and $j$ belong to the same community and 0 otherwise. The goal of community discovery is to maximise $Q$ in order to find the optimal partition of nodes into communities. In the ambiguous vocabulary graph, if two categories are often confused by the model, they are likely to be in the same community.

# 5 EXPERIMENT

## 5.1 DATASETS AND IMPLEMENTATION DETAILS

Following previous OVS works (Cho et al., 2023; Xie et al., 2023; Xu et al., 2023), we train the models on the COCO-Stuff171 (Caesar et al., 2018) dataset with 171 categories and perform zero-shot evaluation on ADE20K (Zhou et al., 2019) and PASCAL-Context (Mottaghi et al., 2014) datasets. ADE20K has two types of annotations namely ADE150 with 150 classes, and ADE847 with 847 classes. PASCAL-Context has the most frequent 59 classes annotation version PC59, and fully annotated version PC459 with 459 categories.

In this work, we utilise the following benchmark models for evaluation: SAN (Cho et al., 2023), CAT-Seg (Cho et al., 2023), SED (Xie et al., 2023), and MAFT+ (Jiao et al., 2024). In addition to the default setting, which uses dataset-specific vocabulary during inference for each dataset, we also created a joint-dataset inference vocabulary set, denoted by $\star$. This joint-dataset vocabulary set is disjoint from PC59, ADE150, PC459, and ADE847, resulting in a total of 1,086 vocabularies. The threshold $\hat{\tau}$ is set to 0.8.

We follow exactly the same configuration for experiments with the benchmark models. The experiments were conducted on a NVIDIA A100 GPU.

## 5.2 RE-BENCHMARKING

Table 1: Quantitative results of our proposed mask-wise evaluation protocol. The symbol $\star$ indicates using a joint-dataset vocabulary set during testing. NULL denotes non-out-matched mask is predicted. Results of the conventional argmax pixel-wise approach are shown in the first four rows.

| Method | Venue | PC59 | | | ADE150 | | | PC459 | | | ADE847 | | |
|---|---|---|---|---|---|---|---|---|---|---|---|---|---|
| SAN | CVPR'23 | 57.70 | | | 32.10 | | | 15.70 | | | 12.40 | | |
| CAT-Seg | CVPR'24 | 63.30 | | | 37.90 | | | 23.80 | | | 16.00 | | |
| SED | CVPR'24 | 60.90 | | | 35.30 | | | 22.10 | | | 13.70 | | |
| MAFT+ | ECCV'24 | 59.40 | | | 36.10 | | | 21.60 | | | 15.10 | | |
| | | front↑ | back↑ | err↓ | front↑ | back↑ | err↓ | front↑ | back↑ | err↓ | front↑ | back↑ | err↓ |
| SAN | CVPR'23 | 65.91 | 93.75 | 9.99 | 42.89 | 93.12 | 8.56 | 27.65 | 70.87 | 6.67 | 22.84 | 92.46 | 8.41 |
| CAT-Seg | CVPR'24 | 68.46 | 94.24 | Null | 45.74 | 94.61 | 5.53 | 30.95 | 68.96 | 3.86 | 26.39 | 93.66 | 5.20 |
| SED | CVPR'24 | 66.29 | 94.21 | 6.43 | 44.90 | 93.50 | 5.20 | 31.41 | 70.72 | 4.93 | 26.99 | 92.61 | 5.07 |
| MAFT+ | ECCV'24 | 64.95 | 93.57 | 9.10 | 46.51 | 93.10 | 7.31 | 31.89 | 70.82 | 7.12 | 28.72 | 92.15 | 7.84 |
| SAN★ | CVPR'23 | 64.32 | 91.83 | 10.99 | 42.18 | 91.50 | 8.32 | 27.85 | 69.06 | 6.20 | 21.01 | 91.04 | 5.10 |
| CAT-Seg★ | CVPR'24 | 66.35 | 92.24 | 2.19 | 50.04 | 92.68 | 2.30 | 11.56 | 67.32 | 2.00 | 12.83 | 91.20 | 2.20 |
| SED★ | CVPR'24 | 63.35 | 91.32 | 5.31 | 42.65 | 91.28 | 4.52 | 30.04 | 68.40 | 3.23 | 27.45 | 90.05 | 4.10 |
| MAFT+★ | ECCV'24 | 62.05 | 91.55 | 8.56 | 44.30 | 91.32 | 6.70 | 29.04 | 69.01 | 4.40 | 26.01 | 90.50 | 6.40 |

Here we compare the commonly used pixel-wise mIoU metric that uses the argmax operation and our proposed mask-wise metric incorporating soft set prediction. The results are shown in Table 1. Our observations are as follows: 1) Simply replacing the pixel-wise argmax-based mIoU with the proposed mask-wise metric, *front* (target), leads to a performance improvement in existing OVS models. The OVS model achieves high accuracy (above 90%) for *back* (non-target) across all datasets, except for the PC459 dataset. 2) As the inference vocabulary increases (using the joint-dataset vocabulary set during testing), our proposed evaluation method maintains relatively stable performance for both *front* and *back*, whereas the performance of the previous argmax-based pixel-wise evaluation method drops significantly, as shown in Table 2. The performance gap here is caused by the ambiguous prediction while they have a high overlap in IoU with ground truth masks. Furthermore, with the increase in the number of inference words, the model's error rate also increases.

# 6 DISCUSSION

## 6.1 ANALYSIS OF MASK-WISE EVALUATION PROTOCOL

To validate the effectiveness of the proposed mask-wise evaluation protocol, we present several key arguments that support its feasibility and practical utility, particularly in the open-world setting.

**The effectiveness.** Previous methods fail to quantitatively measure model performance in open-world settings because they usually rely on category-level matching and evaluate only when the

Table 2: The quantitative results of argmax (*i.e.* pixel-wise) evaluation using a joint-dataset vocabulary set during testing.

| Method | PC59 | ADE150 | PC459 | ADE847 |
|--------|------|--------|-------|--------|
| SAN | 40.15 | 22.50 | 10.50 | 3.20 |
| CAT-Seg | 42.90 | 25.60 | 12.30 | 7.00 |
| SED | 43.70 | 24.10 | 11.00 | 5.20 |
| MAFT+ | 41.30 | 23.80 | 10.80 | 6.50 |

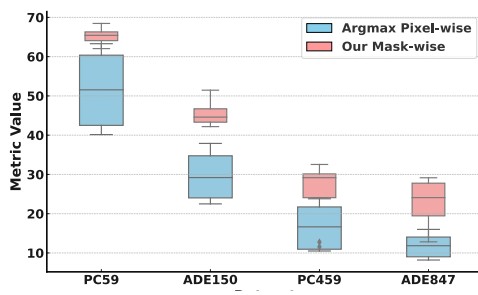

Figure 3: The stability of evaluation methods in comparison when evaluated with more vocabulary (*i.e. in a more open setting*).

predictions belong to a predefined set of categories. This approach often draws incorrect conclusions when faced with new categories or synonyms (such as *"sofa"* vs. *"long couch"*). The quantitative results of previous evaluation methods are compared with those of our method in Table. 3 where the vanilla represents the argmax-based pixel-wise evaluation.

Our proposed evaluation method effectively eliminates the impact of ambiguous category labels via mask matching, focusing on the mask overlap between the predictions and manual annotations. Even in the case of joint datasets, where different datasets use different vocabularies to describe the same or similar categories, our method is still able to evaluate through accurate mask overlap (*e.g.* IoU), avoiding evaluation instability caused by differences in category terms. This mask-based evaluation method allows our scheme to seamlessly adapt to the fusion of multiple datasets without unifying or standardising the category labels of each dataset. In open-world settings, despite the limited number of manually annotated categories, the OVS model is still able to predict as many new concepts as possible, further demonstrating the applicability of the evaluation method.

**The stability.** Our method is evaluated based on mask-wise matching, which means that its main evaluation criterion is the spatial overlap (such as IoU) between the predicted mask and the true annotated mask, rather than relying solely on the matching of class labels. When the number of vocabulary increases, traditional evaluation methods may experience large fluctuations due to the complexity of class label matching, especially when there are synonyms or inter-class ambiguities between different vocabulary sets. However, since our method only relies on matching the geometry and position of the mask, when the vocabulary set expands, the evaluation results will not be significantly affected by the expansion or change of class labels.

Table 3: The quantitative comparison with previous proposed evaluation methods, SG-IoU (Liu et al., 2024) and Open-IoU (Zhou et al., 2023).

| Model | Venue | ADE150 | | | | |
|-------|-------|--------|--------|----------|-------|------|
| | | vanilla | SG-IoU | Open-IoU | front | err |
| SAN | CVPR'23 | 31.88 | 32.92 | 39.00 | 42.89 | 8.56 |
| CAT-Seg | CVPR'24 | 35.68 | 36.75 | 39.90 | 45.74 | 5.53 |
| SED | CVPR'24 | 35.30 | 36.40 | - | 44.90 | 5.20 |
| MAFT+ | ECCV'24 | 36.10 | 37.08 | - | 46.51 | 7.31 |

| Model | Venue | PC459 | | | | |
|-------|-------|--------|--------|----------|-------|------|
| | | vanilla | SG-IoU | Open-IoU | front | err |
| SAN | CVPR'23 | 20.83 | 16.72 | 19.90 | 27.65 | 6.67 |
| CAT-Seg | CVPR'24 | 22.23 | 17.91 | 20.30 | 30.95 | 3.86 |
| SED | CVPR'24 | 22.10 | 18.22 | - | 31.41 | 4.93 |
| MAFT+ | ECCV'24 | 21.60 | 16.45 | - | 31.89 | 7.12 |

| Model | Venue | ADE847 | | | | |
|-------|-------|--------|--------|----------|-------|------|
| | | vanilla | SG-IoU | Open-IoU | front | err |
| SAN | CVPR'23 | 13.07 | 14.17 | 19.20 | 22.84 | 8.41 |
| CAT-Seg | CVPR'24 | 14.53 | 15.64 | 18.40 | 26.39 | 5.20 |
| SED | CVPR'24 | 13.70 | 14.89 | - | 26.99 | 5.07 |
| MAFT+ | ECCV'24 | 15.10 | 16.79 | - | 28.72 | 7.84 |

Assume that we have a universal vocabulary set $\mathcal{V}_{open}$, given a fixed testing dataset with $k$ ground truth categories and corresponding $\mathcal{V}_k$ vocabularies, a suitable vocabulary set $\mathcal{V}_{test}$ for OVS testing on this dataset needs to satisfy: $\mathcal{V}_k \subseteq \mathcal{V}_{test}$. If we set the scale of testing vocabulary set to $N$, there exists $C$ possible subset vocabulary $\mathcal{V}_{sub}$ to satisfy as a good testing vocabulary set for the given dataset, where $C = \frac{\mathcal{V}_{test}!}{(N-k)!(\mathcal{V}_{test}-(N-k))!}$. We then can use the Monte-Carlo method to approximate the upper bound and lower bound of test accuracy (Chen et al., 2019b) given various point estimations

$(ACC_1, ..., ACC_C | \mathcal{V}_{open})$, for both the argmax with pixel-wise mIoU and our mask-wise evaluation as presented in Fig. 3. Specifically, the dataset-specific vocabulary set and joint vocabulary set are just two of many point estimations. We can observe that the proposed mask-wise evaluation protocol maintains a stable measurement across different datasets compared to the commonly used argmax-based and pixel-wise mIoU.

Our method is evaluated based on mask-wise matching, which means that its main evaluation criterion is the spatial overlap (such as IoU) between the predicted mask and the true annotated mask, rather than relying solely on the matching of class labels. When the vocabulary increases, traditional evaluation methods may experience large fluctuations due to the complexity of class label matching, especially when there are synonyms or inter-class ambiguities between different vocabulary sets. However, since our method only relies on matching the geometry and position of the mask, when the vocabulary set expands, the evaluation results will not be significantly affected by the expansion or change of class labels.

Assume that we have a universal vocabulary set $\mathcal{V}_{open}$, given a fixed testing dataset with $k$ ground truth categories and corresponding $\mathcal{V}_k$ vocabularies, a suitable vocabulary set $\mathcal{V}_{test}$ for OVS testing on this dataset needs to satisfy: $\mathcal{V}_k \subseteq \mathcal{V}_{test}$. If we set the scale of testing vocabulary set to $N$, there exists $C$ possible subset vocabulary $\mathcal{V}_{sub}$ to satisfy as a good testing vocabulary set for the given dataset, where:

$$C = \frac{\mathcal{V}_{test}!}{(N-k)!(\mathcal{V}_{test} - (N-k))!} \tag{13}$$

We then can use the Monte-Carlo method to approximate the upper bound and lower bound of test accuracy (Chen et al., 2019b) given various point estimations $(ACC_1, ..., ACC_C | \mathcal{V}_{open})$, for both the argmax with pixel-wise mIoU and our mask-wise evaluation. Specifically, the dataset-specific vocabulary set and joint vocabulary set are just two of many point estimations. We can observe that the proposed mask-wise evaluation protocol maintains a stable measurement across different datasets compared to the commonly used argmax-based and pixel-wise mIoU.

## 6.2 ANALYSIS OF AMBIGUOUS VOCABULARY GRAPH

Using community discovery methods, we can partition the ambiguous graph into communities as shown in Fig. 4a. Each community represents a cluster of classes that are often confused with each other. For example, in an object detection or segmentation dataset, we might observe that the categories *"sofa"*, *"couch"*, and *"armchair"* form a tightly connected community, indicating that these classes are frequently misclassified or confused by the model.

This insight suggests that the dataset may contain ambiguous annotations where these objects are not clearly distinguishable or where multiple terms are used interchangeably in different regions or contexts. By visualising the labels of the same community, as shown in Fig. 4b, we find that

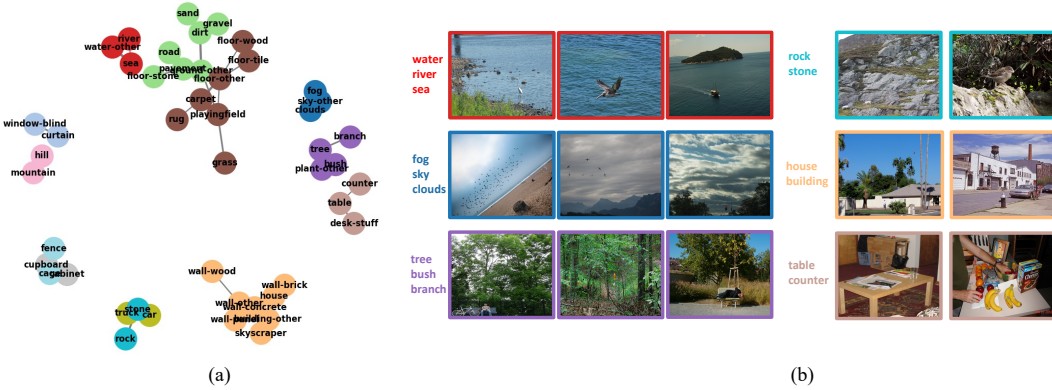

(a)                          (b)

Figure 4: (a) A community extracted from the COCO-Stuff171 dataset (showing only 50 classes). (b) Example images from the same community, where images from the same vocabulary community exhibit visually similar semantics (best viewed in colour).

Table 4: Quantitative results of reducing ambiguous vocabulary through non-target vocabulary removal using our proposed evaluation protocol. Specifically, a proportion $p$ of non-target vocabulary is discarded during training, with $p$ defaulting to $0.9$. The notations "SED w/ 0.7", "SED w/ 0.5", etc., denote results obtained with different values of $p$, where $p = 0.7, 0.5$, etc.

| Method | PC59 | | | ADE150 | | | PC459 | | | ADE847 | | |
|---|---|---|---|---|---|---|---|---|---|---|---|---|
| | front↑ | back↑ | error↓ | front↑ | back↑ | error↓ | front↑ | back↑ | error↓ | front↑ | back↑ | error↓ |
| MAFT+ | -0.87 | -0.23 | -1.98 | -1.56 | -0.63 | -2.30 | -1.12 | -0.08 | -1.24 | -1.81 | -1.33 | -1.96 |
| SED | +2.03 | +0.61 | -2.15 | +5.28 | +0.46 | -0.21 | +0.50 | -0.04 | -1.57 | +1.51 | -0.62 | -2.00 |
| SED w/ 0.7 | +1.10 | +0.40 | -1.80 | +3.40 | +0.30 | -0.70 | +0.25 | -0.02 | -1.50 | +1.00 | -0.30 | -1.80 |
| SED w/ 0.5 | +1.40 | +0.50 | -1.70 | +3.90 | +0.40 | -0.60 | +0.35 | -0.01 | -1.40 | +1.30 | -0.40 | -1.70 |
| SED w/ 0.3 | +1.70 | +0.55 | -1.60 | +4.20 | +0.42 | -0.50 | +0.40 | +0.01 | -1.30 | +1.40 | -0.50 | -1.60 |
| SED w/ 0.1 | +2.00 | +0.60 | -1.50 | +4.50 | +0.45 | -0.40 | +0.45 | +0.02 | -1.20 | +1.50 | -0.55 | -1.50 |

these labels are extremely similar visually and difficult to distinguish through subtle visual differences. From a human perspective, these labels are likely to be classified as the same thing, showing extremely high similarity, which may indicate that they share some core features or attributes.

Additionally, the community discovery process helps reveal systematic biases in the dataset. For example, if certain categories (such as "sedan" "SUV" and "truck") are often clustered in the same community, this may indicate that the dataset lacks diversity in the representation of different vehicle types, or that the model is inadequate in distinguishing between these categories. Details of the community discovery results for the datasets used in this paper can be found in Appendix A.2.

Experiments show that reducing ambiguous vocabulary helps the one-stage model focus on meaningful semantic distinctions and avoid distractions from subtle differences between similar categories, as shown in Table 4. This non-target vocabulary removal strategy aligns the language likelihood more closely with real-world distributions (Eq. 3), enhancing training efficiency. Since under real-world scenario, vocabulary distributions are inherently uncertain and often include out-of-distribution vocabularies absent from the training dataset. By dynamically perturbating the vocabulary set by randomly discarding non-target vocabularies during training, the model is exposed to diverse vocabulary subsets, mitigating reliance on a fixed vocabulary distribution. This perturbation promotes better generalization and adaptability to unseen word distributions, ultimately strengthening the model's robustness in open-world scenarios.

## 6.3 SUMMARY OF THE OBSERVATIONS

Based on the discussion above, we summarise the key findings as follows: 1) Through the analysis of our ambiguous vocabulary graph, we identified the presence of numerous ambiguous or synonymous vocabularies in commonly used OVS datasets. 2) After conducting community discovery analysis on the ambiguous vocabulary graph established from the model's predictions, we found that the categories the model tends to confuse often belong to the same community, and their corresponding images are visually similar. 3) We further proposed to remove such ambiguous vocabularies during the training stage, by simply randomly discarding non-target vocabularies, and found it led to performance improvements for the OVS model.

## 7 CONCLUSION

In conclusion, in this paper, we introduced a mask-wise evaluation protocol for Open-Vocabulary Segmentation (OVS) to address the issue of ambiguous vocabulary in evaluation. These ambiguities often arise under open-world conditions, where multiple interpretations of labels can be valid. Our experiments validated the effectiveness of the proposed evaluation approach. Moreover, using our evaluation protocol, we can construct an ambiguous vocabulary graph for OVS models, revealing a significant presence of confusing annotations in current OVS datasets. The experiments further showed that reducing such ambiguities can enhance the generalisation capability of OVS models, leading to improved performance. In addition, a further discussion provided insights for follow-up research. We hope our study could encourage the community to think more about the openness of open-world problems and hopefully inspire new research questions.

## 8 ACKNOWLEDGEMENTS

This project is partially supported by the Royal Society grants (SIF\R1\231009, IES\R3\223050) and an Amazon Research Award. Qiming Huang is supported by the China Scholarship Council (Grant No. 202408060321). The computations in this research were performed using the Baskerville Tier 2 HPC service. Baskerville was funded by the EPSRC and UKRI through the World Class Labs scheme (EP\T022221\1) and the Digital Research Infrastructure programme (EP\W032244\1) and is operated by Advanced Research Computing at the University of Birmingham.

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

# A APPENDIX

## A.1 PSEUDOCODE OF THE MASK-WISE EVALUATION PROTOCOL

---
**Algorithm 1** Mask-Wise Evaluation Protocol

---
**Require:** Annotated masks list $\{\mathbf{M}_i^{true}\}$, predicted masks list $\{\mathbf{B}(\tau)_i\}$, threshold range $\{0.1, 0.2, \ldots, 0.9\}$, and threshold $\tau_{AV}$ for ambiguous vocabulary matrix.
**Ensure:** Results for each threshold $\tau$: $\{ \mathbf{CM}_\tau, \mathbf{EM}_\tau, \mathbf{AV}_\tau \}$
 1: Initialize results dictionary **Results** $\leftarrow \emptyset$
 2: **for** $\tau \in \{0.1, 0.2, \ldots, 0.8, 0.9\}$ **do**         ▷ Iterate over all threshold values
 3:      Initialize confusion matrix $\mathbf{CM}_\tau \leftarrow 0$
 4:      Initialize ambiguous vocabulary matrix $\mathbf{AV}_\tau \leftarrow 0$
 5:      Initialize error matrix $\mathbf{EM}_\tau \leftarrow 0$
 6:      **for** $i \in \mathcal{D}$ **do**         ▷ Iterate over all images in the dataset
 7:          Obtain binary predicted masks $\mathbf{B}(\tau)_i$
 8:          **for** $c \in C^{\text{true}}$ **do**         ▷ Iterate over true categories
 9:             Compute binary confusion matrix for $\mathbf{M}_{i,c}^{true}$ and $\mathbf{B}(\tau)_{i,c}$
10:             Update confusion matrix $\mathbf{CM}_{\tau,c}$ with computed values
11:          **end for**
12:          Remove masks not in $C^{\text{true}}$ and zero masks from predictions:

$$\mathbf{B}(\tau)_i^{\text{rem}} \leftarrow \{\mathbf{B}(\tau)_{i,c} \mid c \notin C^{\text{true}}, \mathbf{B}(\tau)_{i,c} \neq 0\}$$

13:          Solve bipartite graph matching between $\mathbf{B}(\tau)_i^{\text{rem}}$ and $\mathbf{M}_i^{\text{true}}$:

$$\text{Matches} \leftarrow \{(b, m) \mid b \in \mathbf{B}(\tau)_i^{\text{rem}}, m \in \mathbf{M}_i^{\text{true}}, \text{IoU}(b, m) > \tau_{AV}\}$$

14:          **for** each matched pair $(b, m) \in$ Matches **do**
15:             Update $\mathbf{AV}_{\tau,m,b} \leftarrow \mathbf{AV}_{\tau,m,b} + 1$
16:             Remove matched mask $b$ from $\mathbf{B}(\tau)_i^{\text{rem}}$
17:          **end for**
18:          Let $\hat{\mathbf{B}}(\tau)_i^{\text{rem}}$ be the remaining masks after removal
19:          **for** $c \in$ Categories in $\hat{\mathbf{B}}(\tau)_i^{\text{rem}}$ **do**         ▷ Iterate over remaining categories
20:             Compute error $err_{i,c}$ for image $i$
21:             Update error matrix $\mathbf{EM}_{\tau,c} \leftarrow \mathbf{EM}_{\tau,c} + err_{i,c}$
22:          **end for**
23:      **end for**
24:      Store results for current $\tau$:

$$\textbf{Results}[\tau] \leftarrow \{\mathbf{CM}_\tau, \mathbf{AV}_\tau, \mathbf{EM}_\tau\}$$

25: **end for**
26: **return Results**

---

## A.2 THE COMMUNITY DISCOVERY RESULTS ACROSS THE DATASETS.

COCO-Stuff171

```
({'clouds', 'fog', 'sky-other'},
 {'building-other',
  'curtain',
  'house',
  'skyscraper',
  'wall-brick',
  'wall-concrete',
  'wall-other',
  'wall-panel',
  'wall-stone',
  'wall-tile',
  'wall-wood',
  'window-blind',
  'window-other',
  'wood'},
 {'carpet',
  'dirt',
  'floor-marble',
  'floor-other',
  'floor-stone',
  'floor-tile',
  'floor-wood',
  'gravel',
  'ground-other',
  'pavement',
  'platform',
  'playingfield',
  'road',
  'rug',
  'sand'},
 {'river', 'sea', 'water-other'},
 {'branch',
  'bush',
  'flower',
  'grass',
  'hill',
  'mountain',
  'plant-other',
  'potted plant',
  'straw',
  'tree'},
 {'cabinet', 'cupboard', 'shelf'},
 {'bus', 'car', 'train', 'truck'},
 {'rock', 'stone'},
 {'cage', 'fence', 'railing', 'structural-other'},
 {'counter', 'desk-stuff', 'dining table', 'table'},
 {'cloth', 'textile-other'},
 {'clothes', 'person'},
 {'furniture-other', 'metal', 'plastic', 'stop sign'},
 {'backpack', 'handbag'},
 {'cup', 'wine glass'},
 {'ceiling-other', 'roof'},
 {'hot dog', 'sandwich'})
```

PC59

```
({'bench',
  'building',
  'cabinet',
  'ceiling',
  'chair',
```

```
 'curtain',
 'diningtable',
 'door',
 'fence',
 'floor',
 'flower',
 'grass',
 'ground',
 'mountain',
 'platform',
 'pottedplant',
 'road',
 'rock',
 'shelves',
 'sidewalk',
 'track',
 'tree',
 'wall',
 'window',
 'wood'},
{'bus', 'car', 'truck'},
{'bag', 'bed', 'bedclothes', 'cloth', 'dog', 'sofa'},
{'boat', 'water'},
{'computer', 'tvmonitor'},
{'bicycle', 'motorbike'})
```

ADE150

```
({'awning',
 'blind',
 'booth',
 'building',
 'ceiling',
 'chandelier',
 'curtain',
 'door',
 'fence',
 'hill',
 'house',
 'lamp',
 'light',
 'mountain',
 'railing',
 'rock',
 'sconce',
 'screen door',
 'skyscraper',
 'tower',
 'wall',
 'windowpane'},
{'canopy',
 'dirt track',
 'earth',
 'field',
 'floor',
 'flower',
 'grass',
 'land',
 'path',
 'plant',
 'road',
 'rug',
 'runway',
 'sand',
 'sidewalk',
```

```
 'tree'},
{'cushion', 'pillow'},
{'car', 'truck', 'van'},
{'armchair', 'chair', 'seat', 'sofa', 'stool', 'swivel chair'},
{'bookcase',
 'cabinet',
 'chest of drawers',
 'coffee table',
 'counter',
 'countertop',
 'desk',
 'kitchen island',
 'pool table',
 'shelf',
 'table'},
{'oven', 'stove'},
{'lake', 'river', 'sea', 'water'},
{'crt screen', 'monitor', 'television receiver'},
{'ashcan', 'pot', 'vase'},
{'stairs', 'stairway'},
{'poster', 'signboard'})
```

PC459

```
({'brick',
 'bridge',
 'building',
 'cabinet',
 'cabinetdoor',
 'cage',
 'ceiling',
 'closet',
 'concrete',
 'counter',
 'door',
 'fence',
 'floor',
 'footbridge',
 'ground',
 'handrail',
 'mat',
 'metal',
 'patio',
 'platform',
 'pole',
 'road',
 'rug',
 'sand',
 'shed',
 'shelves',
 'sidewalk',
 'sign',
 'sky',
 'table',
 'tableware',
 'unknown',
 'wall'},
{'car', 'toycar', 'truck'},
{'light', 'lightbulb'},
{'bag',
 'bedclothes',
 'chair',
 'cloth',
 'clothestree',
 'cushion',
```

```
 'flower',
 'grass',
 'leaves',
 'mountain',
 'pack',
 'pillow',
 'plant',
 'plastic',
 'pot',
 'pottedplant',
 'sofa',
 'stool',
 'straw',
 'towel',
 'tree',
 'wood'},
{'curtain', 'window', 'windowblinds'},
{'rock', 'stone'},
{'dolphin', 'water', 'wharf'},
{'box', 'paperbox'},
{'bicycle', 'tricycle'},
{'beer', 'bottle', 'oxygenbottle'},
{'person', 'player'},
{'coffee', 'cup', 'glass'},
{'screen', 'tvmonitor', 'videogameconsole', 'videoplayer'},
{'picture', 'poster'},
{'bird', 'duck'},
{'rail', 'track'},
{'dog', 'fox'},
{'book', 'paper'})
```

ADE847

```
({'baseboard',
 'central reservation',
 'curb',
 'floor',
 'footpath',
 'mat',
 'path',
 'road',
 'rug',
 'sidewalk',
 'skirting board'},
{'balcony',
 'building',
 'building materials',
 'cabin',
 'first floor',
 'house',
 'pane',
 'porch',
 'shop',
 'shops',
 'skyscraper',
 'street number',
 'windowpane'},
{'cover curtain', 'curtain'},
{'flower',
 'forest',
 'plant',
 'plant pots',
 'pot',
 'tree',
 'trunk',
```

```
 'vase',
 'weeds'},
{'bed', 'beds', 'eiderdown'},
{'door', 'door bars', 'doorframe', 'double door'},
{'buffet',
 'cabinet',
 'chest of drawers',
 'coffee table',
 'desk',
 'table',
 'table cloth',
 'tables',
 'television stand'},
{'booth',
 'brick',
 'hill',
 'mountain',
 'mountain pass',
 'rock',
 'rocky formation',
 'shower room',
 'temple',
 'wall'},
{'apparel', 'dummy', 'person', 'trouser'},
{'car', 'truck', 'van'},
{'cushion', 'pillow'},
{'earth', 'field', 'grass', 'land', 'sand'},
{'counter', 'countertop', 'work surface'},
{'armchair', 'chair', 'rocking chair', 'seat', 'stool', 'swivel chair'},
{'fireplace', 'fireplace utensils'},
{'lake', 'river', 'sea', 'shore', 'water'},
{'awning', 'blind'},
{'sofa', 'sofa bed'},
{'light', 'light bulb'}, {'lamp', 'sconce'},
{'screen', 'television receiver'},
{'ceiling', 'eaves', 'roof'},
{'cooker', 'stove'},
{'clock', 'watch'},
{'games table', 'pool table'},
{'ashcan', 'recycling bin'},
{'barrier', 'fence', 'railing'}, {'stairs', 'step'})
```

