# OpenReview forum: "Revisit the Open Nature of Open Vocabulary Semantic Segmentation"
_ICLR.cc/2025/Conference — ICLR 2025 Poster_

### Official Review · Reviewer_TnMo · 2024-10-31

**Soundness:** 3
**Presentation:** 2
**Contribution:** 2
**Rating:** 6
**Confidence:** 4

**Summary:**

This paper gives a deep observations on open-vocabulary semantic segmentation. To address the ambiguous category issue, the authors propose mask-wise evaluation protocol and a confusion vocabulary graph for open-vocabulary datasets. The experiments validate method defectiveness.

**Strengths:**

1. The paper presents an interesting analysis on the openness of open-vocabulary semantic segmentation.

2. The mask-wise evaluation protocol sounds reasonable.

3. The experiments are conducted on multiple existing methods.

**Weaknesses:**

1. The quality of ambiguous vocabulary graph seems important for performance. Currently, the related experiments are not enough. I think it is better to provide more experiments to verify the quality of ambiguous vocabulary graph.

2. The accuracy for front and back is not very clear. I suggest that the authors give an equation to explain it.

3. The comparison of whether reducing ambiguities during training or not is necessary.

**Questions:**

Please refer to weakness.  It is important to give more experiments for ambiguous vocabulary graph and more comparsion.

---

> ### Author Response · Authors · 2024-11-25
> **Response to Reviewer TnMo (part 1)**
>
> # Weakness 1
>
> Thank you for your comment. We clarify that the ambiguous vocabulary graph does not affect model performance or evaluation metrics. It is a tool to analyse overlaps in masks with differing category labels between model predictions and human annotations. While it aids in understanding ambiguous predictions, it is not a factor in performance evaluation, which relies on metrics like $front$, $back$, and $err$. And we provide extra following analysis of the ambiguous vocabulary graph.
>
> We take CAT-Seg model as an example, the top10 ambiguous vocabularies are shown below:
>
> | Rank | COCO (Pair ↔ Frequency)         | PC59 (Pair ↔ Frequency) | ADE150 (Pair ↔ Frequency) | PC459 (Pair ↔ Frequency)   | ADE847 (Pair ↔ Frequency)  |
> | ---- | ------------------------------- | ----------------------- | ------------------------- | -------------------------- | -------------------------- |
> | 1    | Clouds ↔ Sky-other: 520         | Road ↔ Ground: 1082     | House ↔ Building: 157     | Road ↔ Ground: 967         | Road ↔ Ground: 967         |
> | 2    | Sky-other ↔ Clouds: 464         | Sidewalk ↔ Ground: 475  | Rug ↔ Floor: 140          | Sand ↔ Ground: 482         | Sand ↔ Ground: 482         |
> | 3    | Sky-other ↔ Fog: 236            | Wall ↔ Building: 423    | River ↔ Water: 70         | Rug ↔ Floor: 371           | Rug ↔ Floor: 371           |
> | 4    | Road ↔ Pavement: 181            | Floor ↔ Ground: 236     | Skyscraper ↔ Building: 67 | Sidewalk ↔ Ground: 360     | Sidewalk ↔ Ground: 360     |
> | 5    | Wall-other ↔ Wall-concrete: 167 | Truck ↔ Car: 176        | Sidewalk ↔ Road: 58       | Frame ↔ Picture: 247       | Frame ↔ Picture: 247       |
>
> This analysis highlights dataset labelling issues (e.g., ambiguous category definitions) and model limitations (e.g., difficulty in capturing category boundaries), guiding improvements in labelling and model optimisation. For instance, in response to Weakness 3, we enhanced performance by merging ambiguous vocabularies in the training data.
>
> We would be happy to provide further details and experiments if the reviewer have any questions, thanks!
>
> # Weakness 2
>
> Thank you for the comments. The updated Algorithm 1 in the revised version illustrates the process of obtaining the confusion matrix (CM), ambiguous vocabulary matrix (AV), and error matrix (EM). The computation of our three metrics is detailed in Section 4, as shown below:
>
> $front_\tau = \frac{1}{|C|}\sum_{c \in C} \frac{CM_\tau[TP_c]}{CM_{\tau[TP_c]} + CM_\tau[FP_c] + CM_\tau[FN_c]}$
>
> $back_\tau = \frac{1}{|C|}\sum_{c \in C} \frac{CM_\tau[TN_c]}{CM_\tau[TN_c] + CM_\tau[FP_c] + CM_\tau[FN_c]}$
>
> $err_\tau = \frac{1}{|C|}\sum_{c \in C} EM_{\tau, c}$
>
> where $\tau$ is the threshold, $c$ is the class.
>
> We would be happy to provide further details and experiments if the reviewer have any questions, thanks!

---

> ### Author Response · Authors · 2024-11-25
> **Response to Reviewer TnMo (part 2)**
>
> # Weakness 3
>
> As suggested, we further provided the extra comparison of whether reducing ambiguities during training or not is necessary.  (1) The vocabulary dropout is a strategy that **implicitly** reduces ambiguous vocabulary during training. Alongside the Table 4 presents an ablation study on different dropout rates \( p \), reflecting varying levels of ambiguity reduction. We conduct another experiment with the CAT-Seg model as shown below:
>
> |                  |       | PC59  |      |       | ADE150 |      |       | PC459 |      |       | ADE847 |      |
> |------------------|-------|-------|------|-------|--------|------|-------|-------|------|-------|--------|------|
> |                  | front | back  | err  | front | back   | err  | front | back  | err  | front | back   | err  |
> | CAT-Seg Original | 65.93 | 94.2  | 7.0  | 45.74 | 94.07  | 5.53 | 30.95 | 71.37 | 3.86 | 26.39 | 92.85  | 5.20 |
> | CAT-Seg VD 0.1   | 67.27 | 94.38 | 4.32 | 47.62 | 93.88  | 4.11 | 33.21 | 71.74 | 3.71 | 30.64 | 92.55  | 5.02 |
> | CAT-Seg VD 0.3   | 66.81 | 94.32 | 4.01 | 48.15 | 93.88  | 4.64 | 34.17 | 71.7  | 4.44 | 29.42 | 93.41  | 4.22 |
> | CAT-Seg VD 0.5   | 67.31 | 94.36 | 4.22 | 48.32 | 94.13  | 3.96 | 34.85 | 71.68 | 4.61 | 30.49 | 93.18  | 4.57 |
>
> Reducing ambiguous vocabularies during training enhances the zero-shot capability of OVS. Additionally, we manually merged certain vocabularies from the top 10 ambiguous terms in COCO (e.g., *house* and *building* into *house, building*), **explicitly** reducing ambiguity. Training the CAT-Seg model with this refined vocabulary shows a slight improvement, as detailed below:
>
>
> |               | ADE150 | ADE150 | PC459 | PC459 | ADE847 | ADE847 |
> |---------------|--------|--------|-------|-------|--------|--------|
> | Training data | front  | err    | front | err   | front  | err    |
> | COCO          | 45.74  | 5.53   | 30.95 | 3.86  | 26.39  | 5.20   |
> | COCO-Merge    | 46.75  | 5.02   | 31.75 | 3.45  | 27.02  | 4.89   |
>
> However, we would like to emphasise that this is not the main contribution of our paper. Our primary focus is to propose a new evaluation method for OVS models that resolves the issue that traditional image segmentation metrics may misclassify visually similar objects as errors. Ambiguous vocabulary graph (matrix) is a tool helpful for understanding the OVS model predictions.
>
> We would be happy to provide further details and experiments if the reviewer have any questions, thanks!

---

> > ### Comment · Reviewer_TnMo · 2024-12-01
> >
> > About weakness 3, which method is used, SED or CAT-Seg. In text, it is SED, while it is CAT-Seg in Table.

---

> > > ### Author Response · Authors · 2024-12-01
> > >
> > > Thank you for your response. The text in our reply should be "CAT-Seg," and we have made the correction. If you have any further questions, please let us know.

---

### Official Review · Reviewer_tdt4 · 2024-11-01

**Soundness:** 3
**Presentation:** 3
**Contribution:** 2
**Rating:** 8
**Confidence:** 3

**Summary:**

The performance of Open Vocabulary Segmentation (OVS) models will decrease as the query vocabulary size increases, especially when semantically similar category names are present, contradicting the original purpose of OVS. To address this, the authors proposed a mask-wise evaluation protocol based on match/mismatch between prediction and annotation mask pairs, avoiding forced category matching. Key innovations include reducing ambiguity and constructing an ambiguous vocabulary graph. Comprehensive experiments and analysis reveal numerous ambiguous categories in current OVS datasets. Utilizing the proposed protocols during the training and testing stages can help to improve the model’s zero-shot inference capability.

**Strengths:**

1. Good motivation, authors pointed out the current OVS evaluation sets have many semantic similar categories, which may influence the training&testing stages of model, which further influence the inference ability of current OVS methods. Based on this, authors proposed a new evaluation protocols to alleviate this issue.

2. The whole paper is relatively clear and easy to follow.

3. Very comprehensive experiment results on multiple datasets and multiple OVS methods.

**Weaknesses:**

Writing Suggestions:

1. In the Abstract, authors claim that OVS models perform better under the new mask-wise protocol needs further clarification. To make fair comparisons between the mask-wise and pixel-wise protocols, the authors should add more details about how they determine "better" performance. Providing such details would help readers understand the basis for this improvement claim.

2. In the Abstract, the phrase “enhances zero-shot inference capabilities” likely refers to the capabilities of OVS models. Clarifying this would improve readability.

3. Given the similarity between open-vocabulary segmentation and open-vocabulary semantic segmentation, the authors should add a brief section comparing these two concepts. Highlighting key differences in their applications or objectives would help avoid potential confusion and clarify the unique focus of their work.

4. For Equation (5), the authors should provide more detailed motivation for choosing this to determine the best threshold, rather than simply listing the source. It would be helpful if they could explain why this method was selected over alternative approaches and how it specifically benefits their evaluation protocol.

5. The equation at lines 324 to 327 is missing a number.

**Questions:**

1. A significant concern is that the proposed evaluation protocol relies on having sufficient data to identify semantically similar categories. In real-world applications, if the training data lacks adequate masks to differentiate similar categories (e.g., "sofa" and "couch"), the protocol may struggle during testing. To address this, it would be helpful if the authors could analyze the performance of their method with limited training data or provide insights into the minimum data requirements necessary for effective improvement. Additionally, experiments or discussions on the robustness of data scarcity and the impact of potentially misleading information would strengthen the evaluation.


2. While the authors' approach to handling ambiguities through the visual modality is quite interesting, it may be more intuitive to identify similar categories based purely on semantic meaning. For instance, using the text modality to assess semantic similarities could potentially provide greater improvements than relying solely on visual information. To explore this, it would be valuable for the authors to compare their visual-based approach with a text-based semantic similarity approach. Or add more discussions about the potential advantages and disadvantages of incorporating textual semantic information into their method.

---

> ### Author Response · Authors · 2024-11-25
> **Response to Reviewer tdt4 (part 1)**
>
> # Weakness 1
>
> **How to determine "better" performance?** Thank you for the suggestion. In our context, "better" refers to the quantitative superior performance of the OVS model compared to the pixel-wise model within our mask-wise evaluation framework. We attribute this improvement to addressing the limitations of the pixel-wise evaluation framework, where mis-matched categories are forcibly treated as erroneous predictions. We have revised this in the revised version, line 20 - 22. We would be happy to provide further details and experiments if the reviewer have any questions, thanks!
>
> # Weakness 2
>
> Thank you for pointing out the writing suggestions. We intended to refer to the capabilities of OVS models and have corrected this in the revised version, line 25. We would be happy to provide further details and experiments if the reviewer have any questions, thanks!
>
> # Weakness 3
>
> Thank you for pointing out the writing suggestions. "open-vocabulary segmentation" specifically refers to "open-vocabulary semantic segmentation." We have updated all of them in the revised version. We would be happy to provide further details and experiments if the reviewer have any questions, thanks!
>
> # Weakness 4
>
> The best threshold $\tau^\star$ is used to automatically determine the threshold at which the model performs best. It considers simultaneously maximising the value of $front$ and $(1 - err)$. Under our evaluation framework, the theoretical optimal values are $front = 1$ and $err = 0$. This indicates a model that achieves 100% accuracy in identifying target categories while making no incorrect predictions for non-target categories. The best threshold refers to the point closest to the theoretical optimal value in terms of Euclidean distance. We would be happy to provide further details and experiments if the reviewer have any questions, thanks!
>
> # Weakness 5
>
> Thank you for pointing out the writing suggestions. We have updated this in the revised version, line 340.

---

> > ### Author Response · Authors · 2024-11-25
> > **Response to Reviewer tdt4 (part 2)**
> >
> > # Q1
> >
> > Thank you for your valuable feedback. We would like to clarify that our evaluation metrics: $front$, $back$, and $err$ are derived by directly comparing the model's predictions with ground truth annotations and do not depend on large amounts of data or the ability to distinguish semantically similar categories (e.g., "sofa" vs. "couch"). Additionally, we trained CAT-Seg using training datasets with different ratios and reported the evaluation results of our model as shown below. "vanilla" indicates the traditional argmax-based evaluation.
> >
> > |       |         | ADE150 |      |         | PC459 |      |         | ADE847 |      |
> > |-------|---------|--------|------|---------|-------|------|---------|--------|------|
> > | ratio | vanilla | front  | err  | vanilla | front | err  | vanilla | front  | err  |
> > | 100%  |  35.68  |  45.74 | 5.53 |  22.23  | 30.95 | 3.86 |  14.53  |  26.39 | 5.20 |
> > | 50%   |  28.12  |  34.31 | 5.55 |  16.23  | 23.21 | 3.88 |   8.21  |  19.10 | 5.25 |
> > | 10%   |  18.34  |  22.87 | 5.60 |   8.34  | 15.48 | 3.90 |   4.32  |  12.50 | 5.30 |
> >
> > The results show that the front metric, similar to traditional metrics, decreases as the data volume reduces, indicating that the model requires sufficient data to achieve optimal performance. However, the err metric is minimally affected by data volume, suggesting that the ovs model's error predictions are not strongly dependent on the data itself.
> >
> > We would be happy to provide further details and experiments if the reviewer have any questions, thanks!
> >
> > # Q2
> >
> > Thank you for the comments. As suggested, here we compare two text-based semantic similarity approaches, SG-IoU[1] and Open-IoU[2] that incorporating the text-modality during the evaluation stage, shown below.
> >
> > |           |         | **ADE150** |        |          | **PC459** |        |          | **ADE847** |        |          |
> > | --------- | ------- | ---------- | ------ | -------- | --------- | ------ | -------- | ---------- | ------ | -------- |
> > | Model     | venue   | vanilla    | SG-IoU | Open-Iou | vanilla   | SG-IoU | Open-Iou | vanilla    | SG-IoU | Open-Iou |
> > | SAN       | CVPR'23 | 31.88      | 32.92  | 39.00    | 20.83     | 16.72  | 19.90    | 13.07      | 14.17  | 19.2     |
> > | CAT-Seg   | CVPR'24 | 35.68      | 36.75  | 39.90    | 22.23     | 17.91  | 20.30    | 14.53      | 15.64  | 18.40    |
> > | SED       | CVPR'24 | 35.30      | 36.40  | -        | 22.10     | 18.22  | -        | 13.70      | 14.89  | -        |
> > | MAFT-PLUS | ECCV'24 | 36.10      | 37.08  | -        | 21.60     | 16.45  | -        | 15.10      | 16.79  | -        |
> > |           |         |            |        |          |           |        |          |            |        |          |
> > |           |         | **ADE150** |        |          | **PC459** |        |          | **ADE847** |        |          |
> > | Model     | venue   | front      | err    | auc      | front     | err    | auc      | front      | err    | auc      |
> > | SAN       | CVPR'23 | 42.89      | 8.56   | 37.63    | 27.65     | 6.67   | 27.33    | 22.84      | 8.41   | 25.80    |
> > | CAT-Seg   | CVPR'24 | 45.74      | 5.53   | 45.86    | 30.95     | 3.86   | 30.87    | 26.39      | 5.20   | 26.70    |
> > | SED       | CVPR'24 | 44.90      | 5.20   | 45.27    | 31.41     | 4.93   | 31.27    | 26.99      | 5.07   | 28.37    |
> > | MAFT-PLUS | ECCV'24 | 46.51      | 7.31   | 43.24    | 31.89     | 7.12   | 31.65    | 28.72      | 7.84   | 30.42    |
> >
> > However, relying solely on textual similarity presents challenges, such as dependency on the accuracy of text embeddings and difficulties in handling polysemous words within context (e.g., "bat" referring to a flying mammal or a baseball bat). Our method avoids such prerequisites by evaluating solely based on visual masks.
> >
> > Its effectiveness lies in overcoming the limitations of forced category matching. The above experiments have been incorporated into the revised version.
> >
> > We would be happy to provide further details and experiments if the reviewer have any questions, thanks!
> >
> > [1] Open-vocabulary segmentation with semantic-assisted calibration. CVPR 2024
> >
> > [2] Rethinking evaluation metrics of open-vocabulary segmentaion. arXiv preprint, 2023

---

> > > ### Comment · Reviewer_tdt4 · 2024-11-28
> > > **Thanks**
> > >
> > > Thank you for your detailed responses and clarifications. I greatly appreciate your efforts. Kindly incorporate the modified content into the main paper where possible (especially regarding the discussion & comparison of visual modality and text modality parts). Since the authors have addressed most of my concerns, I will increase my rating.

---

> > > > ### Author Response · Authors · 2024-11-28
> > > >
> > > > Thank you, we appreciate your confirmation! It is great to know that we addressed the concerns.
> > > > And yes, we will incorporate the modified content into the main paper.

---

### Official Review · Reviewer_eu9j · 2024-11-03

**Soundness:** 3
**Presentation:** 2
**Contribution:** 2
**Rating:** 5
**Confidence:** 4

**Summary:**

This study proposes new evaluation metrics for Open-Vocabulary Segmentation (OVS) tasks. A key limitation of evaluating OVS methods on fixed-category datasets is that traditional image segmentation metrics may misclassify visually similar objects as errors, even when they are semantically related but belong to different categories. This issue intensifies with an increasing number of category labels in the test dataset. This issue becomes more pronounced as the number of category labels in the test data increases. Previous research has addressed this challenge, resulting in improved metrics such as Open-mIoU and SG-IOU. The central premise of this work is to focus evaluation on mask similarity rather than textual similarity.

**Strengths:**

The primary contention of this manuscript is to shift the focus of evaluation from textual to mask similarity in assessing OVS models. The authors have identified a gap in the current assessment metrics, which are deemed inadequate for evaluating OVS models, and have proposed a novel metric to address this issue.

**Weaknesses:**

The manuscript exhibits a lack of clarity and organization in its writing.

**Questions:**

Q1: The analysis in Section 3 appears disconnected from subsequent sections.

Q2: In Figure 2, $\mathbb{A}$ represents a set of predicted binary masks. How are the predicted masks in $\mathbb{B}$ and $\mathbb{C}$ derived from $\mathbb{A}$? If they are matched to GT masks based on IoU using bipartite matching, it seems Figure 2 suggests that the number of predicted masks by the model exceeds that of the ground truth, which is not realistic. Additionally, predicted masks in $\mathbb{B}$ and $\mathbb{C}$ should not overlap according to $\mathbb{C} = \mathbb{A} \backslash \mathbb{B}$.

Q3: The correlation between Algorithm 1 and Section 4 is weak: For example, (1) The CM is not referenced outside the Algorthm 1. (2) The calculations for the core evaluation metrics -- front, back, and errors -- are not represented in Algorithm 1 or any other equations. (3) How is the best threshold $\tau^*$ used in Algorithm 1?

Q4: What constitutes a good evaluation metric? The last sentence of the introduction (line 83 on page 2) implies that the authors equate higher performance values with better evaluation metrics, which is unreasonable.
In Figure 3, the authors seem to suggest that more stable evaluation metrics are preferable; however, this should also be compared with other metrics like Open-mIoU and SG-IoU.

---

> ### Author Response · Authors · 2024-11-25
> **Response to Reviewer eu9j (part 1)**
>
> # Q1
>
> Thank you for the insightful feedback. We recognise the importance of ensuring a coherent narrative throughout the paper. Specifically, Equation (3) introduces $P(V\mid\Theta)$ showing that the model's parameters need to fit the distribution of the given vocabulary, so ambiguous vocabularies present during training may affect the model's optimisation. This serves as the motivation for the analysis of the ambiguous vocabulary graph and the reducing ambiguous vocabularies during training detailed in Section 6.2. We would be happy to provide further details and experiments if the reviewer have any questions, thanks!
>
> # Q2
>
> Thank you for the comments. Apologies for the typo of omitting the definitions in the manuscript and have updated the following in the revised version.
>
> The $\mathbb{A}$ is all the predicted masks, $\mathbb{B}$ is the mask list where the predicted category by the model aligns with the category annotated by human (not all predicted masks). The $\mathbb{\hat{B}}$ is the set of masks obtained by performing bipartite matching between $(\mathbb{A} \setminus \mathbb{B})$ and the GT, where the IoU of the matched pairs exceeds the threshold $\tau_{AV}$. $\mathbb{C}$ is defined as $\mathbb{C}=\mathbb{A} \setminus (\mathbb{\hat{B}} \cup \mathbb{B})$. We would be happy to provide further details and experiments if the reviewer have any questions, thanks!
>
> # Q3
>
> (1) Thank you for the feedback. The updated Algorithm 1 in the revised version illustrates the process of obtaining the confusion matrix (CM), ambiguous vocabulary matrix (AV), and error matrix (EM). We have added the contents revised version, Section 4.2 where CM is then used to compute the metrics $front$ and $back$.
>
> (2) The computation of our three metrics is also detailed in Section 4, as shown below:
>
> $front_\tau = \frac{1}{|C|}\sum_{c \in C} \frac{CM_\tau[TP_c]}{CM_{\tau[TP_c]} + CM_\tau[FP_c] + CM_\tau[FN_c]}$
>
> $back_\tau = \frac{1}{|C|}\sum_{c \in C} \frac{CM_\tau[TN_c]}{CM_\tau[TN_c] + CM_\tau[FP_c] + CM_\tau[FN_c]}$
>
> $err_\tau = \frac{1}{|C|}\sum_{c \in C} EM_{\tau, c}$
>
> where $\tau$ is the threshold, $c$ is the class. We have added this in the revised version.
>
> (3) The best threshold is not directly applied in Algorithm 1. When we obtain the $front$, $back$, and $error$ metrics under different thresholds $\tau$, the best threshold is identified as the threshold that model performances the best.
>
> We would be happy to provide further details and experiments if the reviewer have any questions, thanks!

---

> > ### Author Response · Authors · 2024-11-25
> > **Response to Reviewer eu9j (part 2)**
> >
> > # Q4
> >
> > As suggested, we compare our evaluation metrics with similarity-based metrics SG-IoU and Open-IoU, as shown in the table below. *Vanilla* represents the standard argmax-based mIoU. Since Open-IoU's similarity matrix is not publicly available, we report results from their paper. In our evaluation, $front$, $back$, and $err$ indicate performance at optimal threshold, while $auc$ is a new metric introduced in the rebuttal, inspired by the ROC-AUC curve, reflecting the area under the curve across all thresholds (visualised in Figure 5 in the appendix, revised version).
> >
> > |           |         | **ADE150** |        |          | **PC459** |        |          | **ADE847** |        |          |
> > |-----------|---------|------------|--------|----------|-----------|--------|----------|------------|--------|----------|
> > | Model     | venue   | vanilla    | SG-IoU | Open-Iou | vanilla   | SG-IoU | Open-Iou | vanilla    | SG-IoU | Open-Iou |
> > | SAN       | CVPR'23 | 31.88      | 32.92  |   39.00  | 20.83     | 16.72  |   19.90  | 13.07      | 14.17  |   19.2   |
> > | CAT-Seg   | CVPR'24 | 35.68      | 36.75  |   39.90  | 22.23     | 17.91  |   20.30  | 14.53      | 15.64  |   18.40  |
> > | SED       | CVPR'24 | 35.30      | 36.40  |     -    | 22.10     | 18.22  |     -    | 13.70      | 14.89  |     -    |
> > | MAFT-PLUS | ECCV'24 | 36.10      | 37.08  |     -    | 21.60     | 16.45  |     -    | 15.10      | 16.79  |     -    |
> > |           |         |            |        |          |           |        |          |            |        |          |
> > |           |         | **ADE150** |        |          | **PC459** |        |          | **ADE847** |        |          |
> > | Model     | venue   | front      | err    |    auc   | front     | err    |    auc   | front      | err    |    auc   |
> > | SAN       | CVPR'23 | 42.89      | 8.56   |   37.63  | 27.65     | 6.67   |   27.33  | 22.84      | 8.41   |   25.80  |
> > | CAT-Seg   | CVPR'24 | 45.74      | 5.53   |   45.86  | 30.95     | 3.86   |   30.87  | 26.39      | 5.20   |   26.70  |
> > | SED       | CVPR'24 | 44.90      | 5.20   |   45.27  | 31.41     | 4.93   |   31.27  | 26.99      | 5.07   |   28.37  |
> > | MAFT-PLUS | ECCV'24 | 46.51      | 7.31   |   43.24  | 31.89     | 7.12   |   31.65  | 28.72      | 7.84   |   30.42  |
> >
> > We agree that the quality of an evaluation metric cannot be determined solely by the magnitude of its values. Traditional methods often misclassify visually similar objects as errors, a problem exacerbated as the inference vocabulary grows (Table 2), leading to an underestimation of OVS model performance. While SG-IoU and Open-IoU attempt to address this issue by incorporating textual similarity, they remain limited in capturing nuanced relationships between visually similar objects. Additionally, as shown in Figure 3, our method also exhibits smaller variance in metrics with increasing inference vocabulary, making it more robust for large-scale open-vocabulary segmentation evaluation.
> >
> > We would be happy to provide further details and experiments if the reviewer have any questions, thanks!

---

> ### Author Response · Authors · 2024-12-02
>
> Dear Reviewer eu9j,
>
> Thank you for your contributions to the reviewing work.
>
> As the deadline for reviewer-author discussion is approaching soon, we kindly ask you to take a look at our responses at your convenience, and let us know if your concerns have been well addressed.
>
> We would be more than happy to provide any additional responses or justifications if you have further concern.
>
> Many thanks and looking forward to hearing from you.

---

### Author Response · Authors · 2024-11-27
**Gentle reminder**

Dear Reviewers,

We thank you for your time and contributions to the review. This is a gentle reminder of our responses, please could you take a look at your convenience and let us know if there are any further questions. Please feel free to post your comments and any potential questions, we would be more than happy to be involved in a discussion and provide any needed further justifications/experiments.

Many thanks!

---

> ### Author Response · Authors · 2024-12-03
>
> Dear reviewers,
>
> We thank you again for your effort in reviewing our paper, and appreciate the constructive comments.
> We have addressed the concerns and updated the paper accordingly.
>
> The deadline for reviewers/authors to post messages is approaching, and we may not be able to see your comments soon. We understand your busy schedule, but it would be appreciated if you could let us know if our responses have addressed your previous concerns, and any further comments or questions you may have, and we will do our best to address them before the deadline.
>
> Thanks again for your feedback and time!

---

### Meta-Review · Area_Chair_JQDZ · 2024-12-24

**Metareview:**

This paper proposes a mask-wise evaluation protocol based on match/mismatch between prediction and annotation mask pairs for open vocabulary segmentation.The motivation of this paper is clear and the experimental results are through and convincing. Although some details about the proposed method need further explanation, the authors' responses have provided these details.

**Additional Comments On Reviewer Discussion:**

The authors have provided detailed responses to the reviewers' comments, although the reviewers do not provide further feedbackes. The AC thinks the main concers have been well addressed.

---

### Decision · Program_Chairs · 2025-01-22

Accept (Poster)